# Specialized impulse conduction pathway in the alligator heart

**Bjarke Jensen[1†], Bastiaan J Boukens[1,2†], Dane A Crossley II[3], Justin Conner[3], Rajiv A Mohan[1], Karel van Duijvenboden[1], Alex V Postma[1,4], Christopher R Gloschat[2], Ruth M Elsey[5], David Sedmera[6], Igor R Efimov[2], Vincent M Christoffels[1]\***

[1]Department of Medical Biology, Heart Failure Research Center, Academic Medical Center, University of Amsterdam, Amsterdam, The Netherlands; [2]Department of Biomedical Engineering, George Washington University, Washington, DC, United States; [3]Department of Biological Sciences, University of North Texas, Denton, United States; [4]Department of Clinical Genetics, Academic Medical Center, University of Amsterdam, Amsterdam, The Netherlands; [5]Rockefeller Wildlife Refuge, Louisiana Department of Wildlife and Fisheries, Grand Chenier, United States; [6]Institute of Anatomy, First Medical Faculty, Charles University, and Institute of Physiology, Czech Academy of Sciences, Prague, Czech Republic

**\*For correspondence:**
v.m.christoffels@amc.uva.nl

[†]These authors contributed equally to this work

**Competing interests:** The authors declare that no competing interests exist.

**Abstract** Mammals and birds have a specialized cardiac atrioventricular conduction system enabling rapid activation of both ventricles. This system may have evolved together with high heart rates to support their endothermic state (warm-bloodedness) and is seemingly lacking in ectothermic vertebrates from which first mammals then birds independently evolved. Here, we studied the conduction system in crocodiles (*Alligator mississippiensis*), the only ectothermic vertebrates with a full ventricular septum. We identified homologues of mammalian conduction system markers (*Tbx3-Tbx5, Scn5a, Gja5, Nppa-Nppb*) and show the presence of a functional atrioventricular bundle. The ventricular Purkinje network, however, was absent and slow ventricular conduction relied on trabecular myocardium, as it does in other ectothermic vertebrates. We propose the evolution of the atrioventricular bundle followed full ventricular septum formation prior to the development of high heart rates and endothermy. In contrast, the evolution of the ventricular Purkinje network is strongly associated with high heart rates and endothermy.
DOI: https://doi.org/10.7554/eLife.32120.001

## Introduction

The muscle of the vertebrate heart initiates and conducts the electrical impulse that initiates contraction. It has regionally distinctive properties, such as conduction velocity, underlying coordinated alternating contraction of atrium and ventricle (*Chi et al., 2008*; *Gillers et al., 2015*; *Keith and Flack, 1907*). In addition, hearts of mammals and birds have specialized atrioventricular conduction system tissues composed of distinctive cardiomyocytes (e.g. glycogen-rich, less developed contractile apparatus, highly conductive) that can be morphologically distinguished from the working myocardium. These components, the atrioventricular bundle, bundle branches and Purkinje fiber network, rapidly conduct and distribute the impulse to the ventricular muscle, coordinating cardiac contractions (*Davies et al., 1952*; *Chiodi and Bortolami, 1967*; *van Weerd and Christoffels, 2016*; *Chuck et al., 2004*; *Rentschler et al., 2001*; *Miquerol et al., 2010*). They enable the high heart rates that set mammals and birds apart from similarly sized ectothermic vertebrates (*Davies and Francis, 1946*; *Lillywhite et al., 1999*). The anatomical and electrophysiological analyses that

**eLife digest** Mammals and birds are referred to as 'warm-blooded' animals, because they maintain a constant high body temperature. This requires a lot of energy, so their bodies need to be well supplied with blood at all times. The hearts of mammals and birds contain two important structures that help them do this. The first is a full wall of muscle – called the ventricular septum – that divides the heart into left and right sides. The second is an electrical circuit made of specialized muscle cells that ensures that the heart beats fast enough by sending rapid electrical signals to the rest of the heart muscle. The circuit contains one group of cells in the ventricular septum, called the bundle of His, and another group termed the Purkinje network.

Reptiles, however, do not maintain high body temperatures and are instead often thought of as 'cold-blooded' animals. The hearts of reptiles do not need to pump blood around the body as quickly and have different structures from warm-blooded animals. For example, most reptile hearts do not have a fully developed ventricular septum. The only exceptions are crocodiles, alligators and their relatives (the 'crocodilians'), which do.

Jensen, Boukens et al. therefore wanted to determine if a crocodilian heart also contained a specialized electrical circuit like those of birds and mammals. Previous studies that attempted to answer this question using only anatomical and electrical methods had yielded ambiguous results. As such, Jensen, Boukens et al. combined these methods with genetic techniques for a more detailed study.

First, the ventricular septum of American alligators, a species of crocodilian, was examined, and found to contain a narrow tissue structure that strongly resembled the bundle of His. Indeed, if this presumptive bundle of His was cut, the electrical circuit was broken. Additional genetic analysis of this structure confirmed that genes similar to those active in the mammalian bundle of His were also switched on in alligators. However, recordings of heart activity showed that heart rates and the spread of electrical signals were both slower in alligators than in warm-blooded animals. This suggests that, although alligators have evolved some specialized muscle cells (in the form of a bundle of His), their electrical circuit is still 'incomplete'. The lack of a Purkinje network, for example, would explain why their heart rates remain slow like other reptiles'.

Together these findings add to the current understanding of how the heart works in different animals with varying requirements for energy and blood flow. Also, since crocodiles and warm-blooded birds both evolved from ancient reptiles, detailed descriptions of their heart structures could shed more light on how warm-bloodedness first developed.

DOI: https://doi.org/10.7554/eLife.32120.002

unequivocally demonstrated the specialized atrioventricular conduction system in mammals and birds, yielded, in contrast, highly divergent interpretations when applied to ectotherms (*Davies et al., 1952*; *Chiodi and Bortolami, 1967*; *Jensen et al., 2012*). Nonetheless, the most prevalent view appears to be that the vertebrate taxa that independently gave rise to mammals and birds, represented by extant amphibians and reptiles, are without a specialized cardiac conduction system (*Tessadori et al., 2012*; *Jensen et al., 2017a*; *Burggren et al., 2017*; *Jensen et al., 2014*, *Supplementary file 1*).

The atrioventricular bundle and its branches in both mammals and birds are located in the ventricular septum (*Davies et al., 1952*). Therefore, it is conceivable that the presence of a ventricular septum, rather than high heart rates, resulted in the development of a specialized atrioventricular conduction pathway. Among ectotherms, only crocodilians (alligators, crocodiles, and gharials) have a full ventricular septum and the electrical activation of the ventricle of the freshwater crocodile has been reported to propagate differently from that of other reptiles (*Jensen et al., 2012*; *Christian and Grigg, 1999*). Early anatomical works suggested that atrioventricular canal myocardium of the American alligator projects onto the ventricular septum (*Greil, 1903*; *Swett, 1923*), but there was no mention of bundle branches and the later investigators that unequivocally show the specialized conduction system in mammals and birds (*Davies and Francis, 1946*), could not find specialized tissues in crocodilians (*Davies et al., 1952*; *Davies et al., 1951*). Here, we investigated the ventricles of the American alligator for the presence of atrioventricular conduction system

components to address the question of their origin. To identify any specialized components in the alligator heart, we analyzed the expression of conserved gene markers for the mammalian and chicken atrioventricular bundle and its branches (*Tbx3*, *Tbx5*, *Scn5a*, *Cntn2*), and for the Purkinje network and its trabecular precursor (*Gja5*, *Cntn2*, *Nppa* and *Nppb*) (*Park and Fishman, 2017*). Furthermore, we assessed impulse conduction patterns to investigate its origin and spread in the ventricles. Because previous anatomical investigations led to contradictory interpretations (*Supplementary file 1*), we use complementary functional and molecular criteria to define 'specialization'. We conclude that the alligator has a functional atrioventricular bundle but lacks a specialized Purkinje network.

## Results

### Atrioventricular conduction pathway in alligator hearts

To investigate the possible presence of a specialized atrioventricular conduction pathway in the crocodilian heart, we made an incision in the dorsal ventricular myocardium near the crux. The crux is the intersection of the atrioventricular sulcus and the dorsal descending coronary artery, and it indicates the position of the atrioventricular node and atrioventricular bundle in the mammalian heart (*Crick et al., 1998*). The dorsal incision disrupted only 10–20% of the atrioventricular junctional myocardium, yet resulted in complete atrioventricular block in three out of three hearts (*Figure 1A–B*, *Figure 1—figure supplement 1*). In three other hearts, extensive ventral and lateral incisions did not result in atrioventricular block (*Figure 1C–D*, *Figure 1—figure supplement 1*). Optical mapping of isolated hearts showed the impulse appearing at the dorsal epicardial surface at the midpoint of the ventricle along the interventricular sulcus, consistent with an origin of the impulse from the atrioventricular bundle (*Figure 1E–G*). On the ventral side of the heart, the activation wave propagated from the ventricular apex toward the major arteries (*Figure 1—figure supplements 2*). These activation patterns resemble those of mammals and birds (*Rentschler et al., 2001*; *Reckova et al., 2003*), but differ from those of the *Anolis* lizard, where epicardial activation starts in the ventricular base region closest to the atria (*Figure 1—figure supplement 2*, [*Jensen et al., 2012*]). Measurements in an opened ventricle revealed the occurrence of first activation near the crest of the septum (*Figure 1H–I*). Together, these data suggest the presence of a functional atrioventricular bundle in the alligator septum crest (*Figure 1I*).

### Molecular characterization of the atrioventricular conduction pathway in alligator hearts

We further characterized the atrioventricular conduction system by determining the alligator homologues of marker genes for specific heart components in mammals and birds. We performed RNA-sequencing of micro-dissected atrium, atrioventricular junction, and ventricles of hearts of embryonic American alligators of Ferguson (*Ferguson, 1985*) stage 27. Transcripts were readily detected for homologues of *Tbx3* in the atrioventricular junction (*Hoogaars et al., 2004*), for *Tbx5* in all samples but lowest in the right ventricle (*Moskowitz et al., 2004*), for cardiac sodium channel Nav1.5 (*Scn5a*), for pan-cardiomyocyte markers cTnT (*Tnnt2*) and cTnI (*Tnni3*), and for gap junction subunit Cx40 (*Gja5*) (*Miquerol et al., 2004*) (*Figure 2*), but not for Nav1.8 (*Scn10a*) (*Remme et al., 2009*). We used *in situ* hybridization to localize expression of these marker genes in the heart. In embryonic American alligator stages 13, 16, and 18, the atrioventricular canal was entirely myocardial (*Tnni3*-positive), revealing a full, undisrupted muscular continuity between the atria and the ventricle. In mammals and birds the atrioventricular bundle can be identified by the expression of *Tbx3* (*Hoogaars et al., 2004*). We found *Tbx3* expression in a small domain from the dorsal atrioventricular canal to the crest of the forming ventricular septum (*Figure 3AB*, *Figure 3—figure supplement 1*). The ventral part of the crest showed very little *Tbx3* expression (*Figure 3A*). This pattern resembles that of *Tbx3* in the atrioventricular node and bundle of mammals and birds. Ventricular expression of *Tbx5* and *Scn5a* was highest in the septal crest (*Figure 3A*), similar to the pattern in mice (*Remme et al., 2009*; *Arnolds et al., 2012*) (*Figure 3A*). The embryonic expression patterns of *Tbx3* and *Tbx5* were similar across crocodilian species indicating evolutionary conservation (*Figure 3—figure supplement 1*).

In hearts of 1-year-old juvenile American alligators the myocardial continuity of the atrioventricular canal and the ventricle, including the ventricular septum, was interrupted by collagen at the

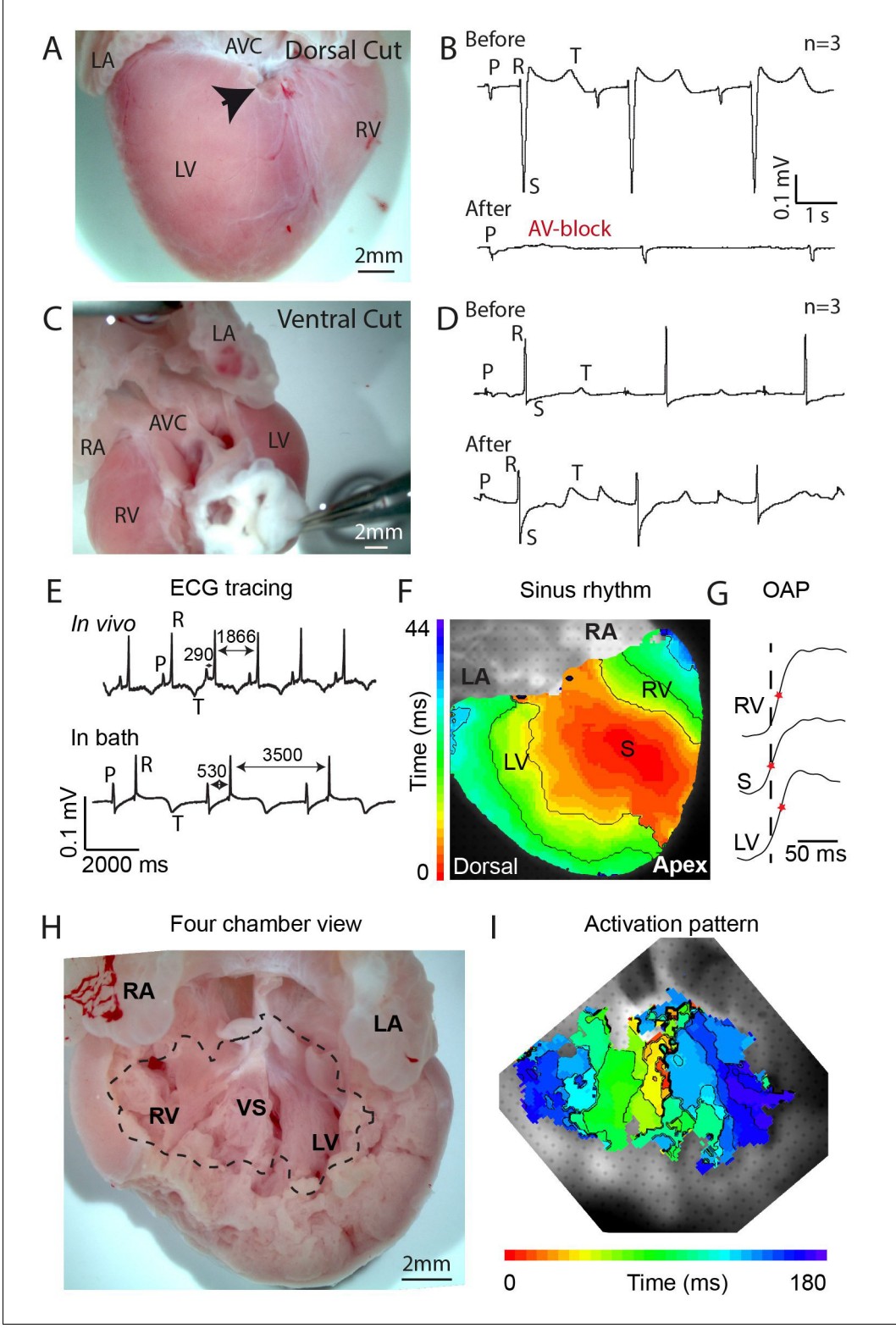

**Figure 1.** Propagation block by cuts in the dorsal alligator atrioventricular canal reveals a specialized atrioventricular conduction system. (A-B) Cuts in a small region of the dorsal atrioventricular canal induces atrioventricular block (dorsal cut n = 3 and ventral cut n = 3). (C-D) Extensive cuts to the ventral and lateral atrioventricular canal do not induce atrioventricular block. (E) *In vivo* and *ex vivo* (in bath) QRS duration was not different (n = 7). (F) In bath optical mapping of ventricular activation revealed epicardial breakthrough of the

*Figure 1 continued on next page*

*Figure 1 continued*

impulse deep in the ventricle (n = 6). (**G**) The maximum rate of depolarization (red star) occurred earlier at position S than at position LV and RV of panel F (n = 6). (**H-I**) Optical recordings from the inside of the heart (**H**) show earliest activation in the septum (n = 1) (**I**). AVC, atrioventricular canal; LA, left atrium; LV, left ventricle; RA, right atrium; RV, right ventricle; s, ventricular sulcus; VS, ventricular septum.

DOI: https://doi.org/10.7554/eLife.32120.003

The following figure supplements are available for figure 1:

**Figure supplement 1.** Overview of cuts (red) to the atrioventricular junction and the extent of collagen (black) in the six juvenile American alligators where atrioventricular conduction was assessed and where optical mapping was performed.

DOI: https://doi.org/10.7554/eLife.32120.004

**Figure supplement 2.** Ventricular activation patterns in reptiles.

DOI: https://doi.org/10.7554/eLife.32120.005

---

ventral and lateral sides (*Figure 1—figure supplement 1*). Dorsally, the atrioventricular canal myocardium was nestled between collagen of the atrioventricular valves and the atrioventricular sulcus (*Figure 4*). The *Tbx3* identified atrioventricular bundle extended from this sheet of atrioventricular canal myocardium (*Figure 4*). Laterally, the *Tbx3* identified atrioventricular bundle was not insulated by connective tissue in contrast to the setting in mammals and birds. *Tbx3* and *Tbx5* were expressed in a pattern similar to that in the embryo, except that *Tbx3* was also expressed deep in the septal crest myocardium (*Figures 3C* and *4*). *Tbx3* was therefore expressed where the earliest electrical activation was found (*Figures 1H–I* and *3C*). *Gja5* was expressed in the *Tbx3*-positive myocardium deep in the ventricular septum, resembling the expression pattern of the atrioventricular bundle in mammals and birds (*Figure 3—figure supplement 2*). Interestingly, the dorsal cuts in alligator hearts inducing atrioventricular block (*Figure 1*) disrupted this *Tbx3*-expressing myocardium (*Figure 3D*).

In mammals and birds, the atrioventricular bundle and Purkinje network are connected by the Tbx3-expressing bundle branches on the left and right flanks of the ventricular septum. We never observed similar branches in the crocodilians, neither with histology nor *in situ* hybridization for *Tbx3*, *Tbx5*, *Gja5*, and *Scn5a*.

## Slowly conducting trabecular wall rather than Purkinje network in alligator hearts

To investigate whether a ventricular Purkinje network was also formed in the crocodilian heart, we set out to collect appropriate markers of the Purkinje network and its embryonic precursor, the trabecular myocardium. In mammals, expression of both *Nppa* and *Nppb* marks the trabecular myocardium during heart development, and *Nppa* expression becomes restricted to the thin subendocardial Purkinje fiber network after birth (*Miquerol et al., 2010*; *Remme et al., 2009*; *Houweling et al., 2005*). *Nppa* and *Nppb* are paralogue genes that are part of the evolutionary ancient natriuretic peptide gene cluster (*Houweling et al., 2005*; *Takei et al., 2011*). Therefore, these genes may be suitable markers to characterize the ventricular conduction system in alligators. However, the archosaur branch, including birds and alligators, have been indicated to have lost *Nppa* during evolution (*Takei et al., 2011*; *Trajanovska and Donald, 2008*). Unexpectedly, we detected the expression of homologues of both *Nppa* and *Nppb* in the RNA-sequencing data of the alligator heart samples (*Figure 5*). Cardiac *Nppb* was expressed strongly, *Nppa* was confined to the atria (*Figure 5*). Sequence homology analysis showed the alligator locus resembles that of turtle and frog, which retained both *Nppa* and *CNP3*, revealing independent divergence of this trabecular myocardium-associated gene cluster in mammals and birds (*Figure 5*).

In embryos of alligator, mouse, and chicken, *Nppb* (and in mouse also *Nppa*) was expressed abundantly in the trabecular myocardium (*Figure 6*, *Figure 6—figure supplement 1*). *Nppb* was absent from the layer expressing *Hey2*, a marker for compact ventricular myocardium in prenatal mammals (*Koibuchi and Chin, 2007*) (*Figure 6B*). One-year juvenile alligators maintained a trabecular layer that expressed *Nppb* (*Figure 6B*). In mammals and birds, *Gja5* expression becomes confined to the Purkinje myocardium (*Miquerol et al., 2010*). In alligator embryos and 1-year juvenile, *Gja5* was expressed not only in the trabeculated myocardium, but also the compact myocardium,

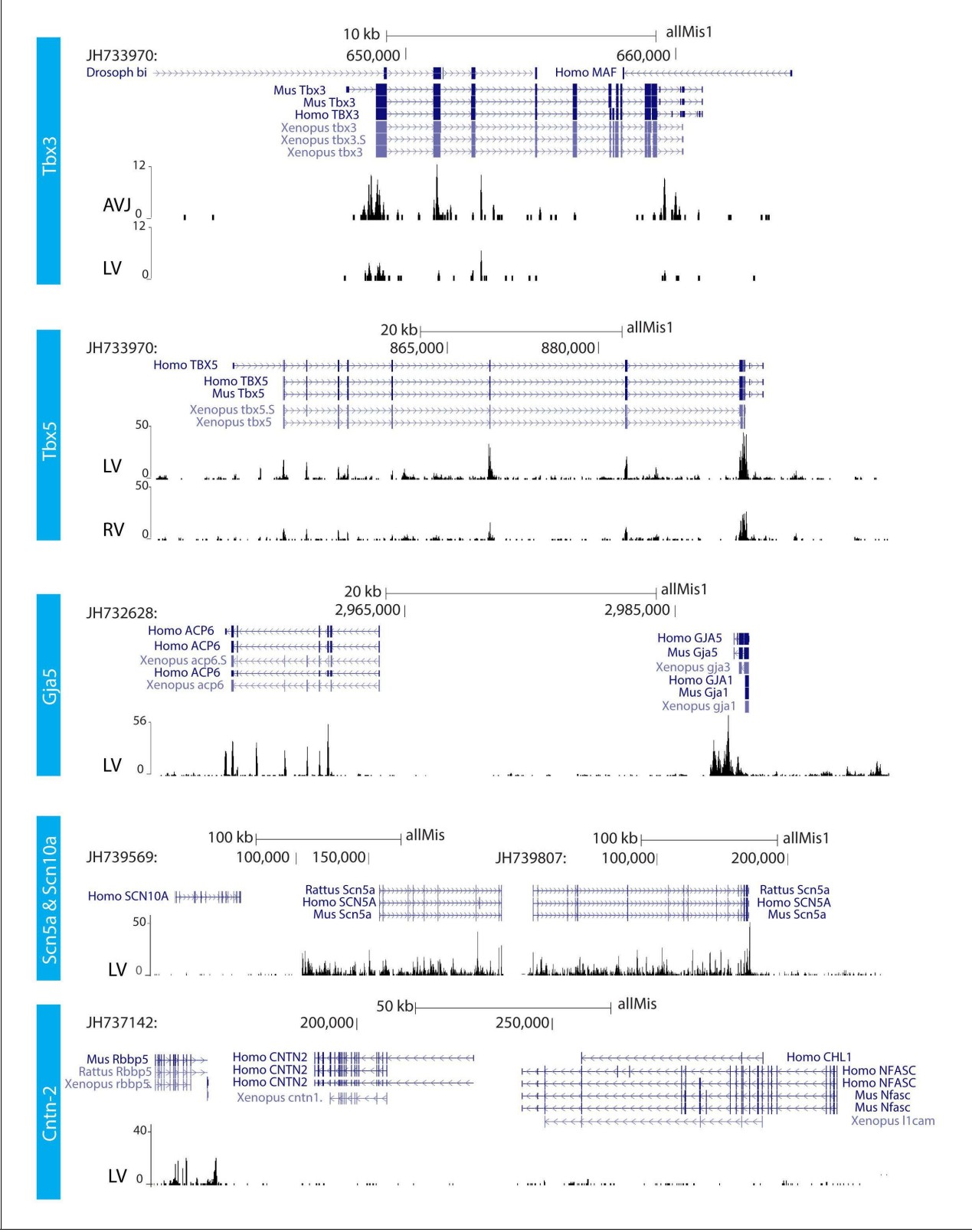

**Figure 2.** RNA sequencing. RNA sequencing of atrioventricular junction (AVJ) and ventricular myocardium of the fetal alligator heart (LV, left ventricle; RV, right ventricle). AVJ sample also contained myocardium from the ventricular base. Note that *Tbx3* tag count was higher in the AVJ compared to the left ventricle. Tag counts of *Tbx5* was higher in the left ventricle than in the right ventricle. Both *Gja5* and *Scn5a* were expressed in the ventricular

*Figure 2 continued on next page*

*Figure 2 continued*

myocardium of the alligator heart. *Cntn2*, however, did not show tag counts indicating absence of expression. (The data are deposited as Jensen B. 2018. Alligator mississippiensis Transcriptome or Gene expression. SRA. https://www.ncbi.nlm.nih.gov/bioproject/PRJNA392860).

DOI: https://doi.org/10.7554/eLife.32120.006

which expressed *Hey2* (*Figure 6—figure supplement 2*). Expression of the mammalian Purkinje network marker *Cntn2* (contactin-2; [*Pallante et al., 2010*]) was not detectable (*Figure 2*). These expression and genetic data indicate that aspects of the trabecular phenotype are maintained in the mature alligator ventricle, as they are in other ectothermic species (*Jensen et al., 2012*), whereas a *Gja5*-positive mammalian/avian-like Purkinje network does not appear to form.

Optical mapping analysis indicated the presence of two functional layers. In two out of six animals, we observed action potentials with fractionated (biphasic) upstrokes, in which an initial steep deflection was followed by a lessening of the inclination only to be followed by a second steep deflection (*Figure 7A–C*, *Figure 7—figure supplement 1*). At both the dorsal and ventral side, we observed the first phase of the upstroke in the vicinity of the ventricular septum. Distinct activation maps could be generated from both the first and the second part of the biphasic upstrokes showing first activation in the mid of the ventricle at the dorsal side (*Figure 7—figure supplement 1*). The dorsal activation patterns based on the first part of the upstroke resembled the dorsal activation patterns of the four animals that did not show fractionated upstrokes. The biphasic upstroke became monophasic at the location near the electrode during ventricular stimulation indicating that each phase of the upstroke corresponded to impulse conduction through a different myocardial layer. To validate this hypothesis, we recorded local electrograms, which showed biphasic deflections at exactly the same regions where we recorded fractionated upstrokes (n = 2) (*Figure 7D*). Together with the gene expression data, these data suggest the presence of two layers, a trabecular myocardial layer and a compact layer, which are activated in subsequent order.

Consistent with the notion of the absence in alligator of a Purkinje network, which in mammals and birds exhibits rapid propagation of the electrical impulse, QRS duration was long in the alligator indicating slow ventricular impulse propagation (110 ± 13 ms, as in previous studies [*Heaton-Jones and King, 1994*]). Temperature affects propagation speed (*Smeets et al., 1986*) and we tested whether alligators at mammalian body temperatures would have a mammal-like QRS duration. Even when heated to mammalian body temperatures, the alligator QRS duration (90 ± 10 ms) was much longer than QRS duration of similarly sized mammals (approximately 30 ms (*Detweiler, 2010*); *Figure 7E–F*).

## Discussion

Here we investigated the atrioventricular conduction system in the crocodlians, the only ectotherms with a ventricular septum. Our results reveal the presence of a specialized conduction pathway connecting the atria with the ventricular septum, that is, an atrioventricular bundle, that is functionally and genetically comparable to that of mammals and birds. We found no indications of the presence of bundle branches. The ventricular conduction system component and the associated genetic wiring of alligators is similar to that of other ectothermic vertebrates, but unlike the Purkinje system in mammals and birds.

Longstanding as well as recent literature has shown that in hearts of ectotherms a non-insulated muscular atrioventricular junction delays and propagates the impulse from the atria to the ventricular base in the absence of anatomically specialized conduction system parts (*Jensen et al., 2012*; *AlanisAlanís et al., 1973*). This resembles the situation in the embryonic heart of both mammals and birds. While some reports have claimed to have observed specialized atrioventricular bundles components in non-crocodilian reptiles on the basis of anatomy, the large majority of studies do not support these claims (*Supplementary file 1*). The presence of an atrioventricular bundle in an ectotherm, as our data indicates, is incompatible with our current understanding of the evolution of the conduction system.

Based on the comprehensive anatomical studies in the first half of the 20th century, it has been concluded that specialized atrioventricular conduction systems are an adaptation to high heart rates and are therefore only found in the endothermic mammals and birds (*Davies et al., 1952*;

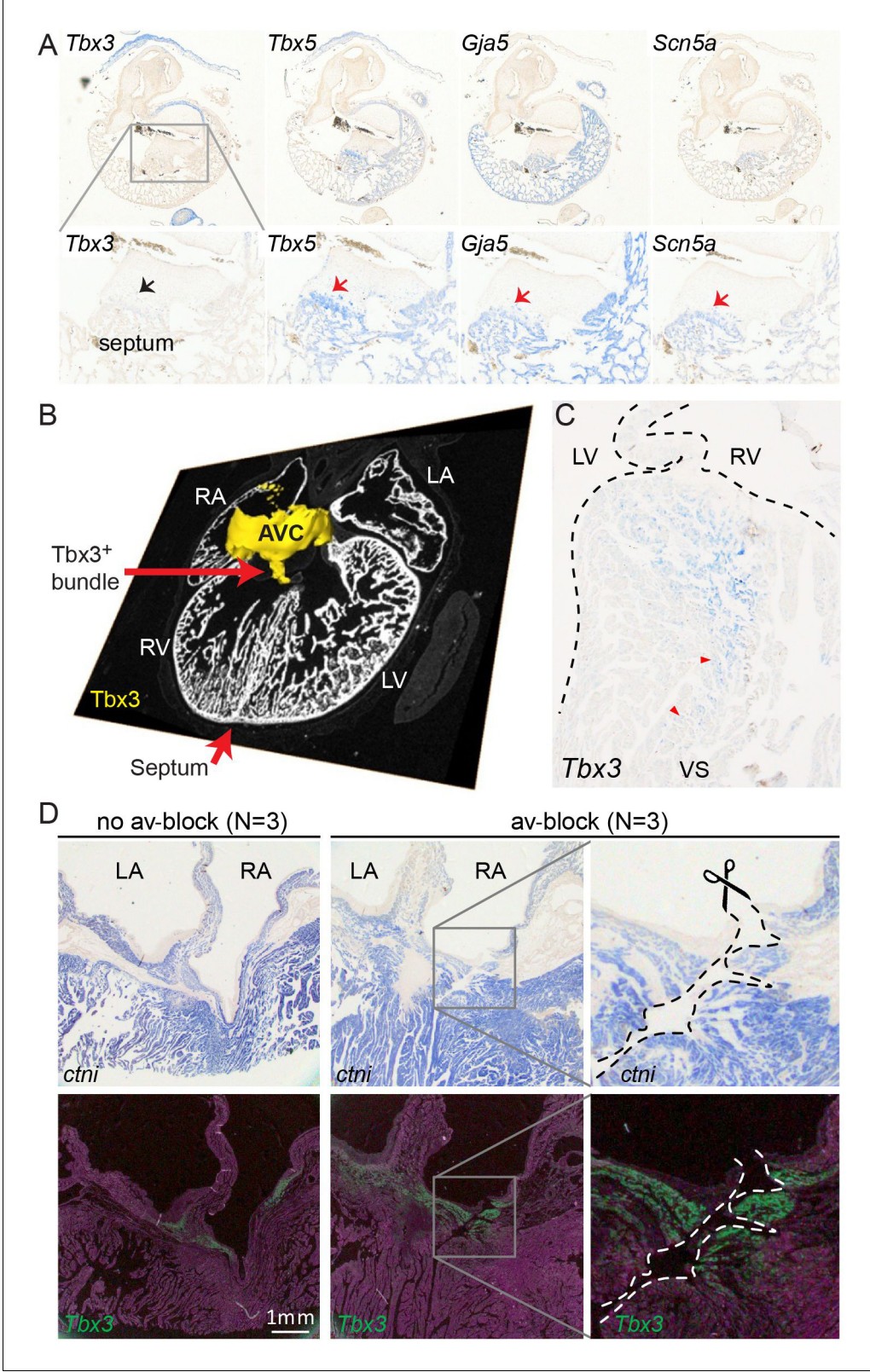

**Figure 3.** Molecular identification of the alligator atrioventricular conduction axis. (**A**) mRNA detection by *in situ* hybridization in the Ferguson stage 16 embryo, showing a mammal and avian like phenotype of the septal crest and trabeculated myocardium. (**B**) Reconstruction of *Tbx3* expression in the Ferguson stage 18 embryo reveals a dorsal bundle connecting the atrioventricular junction to the ventricular septum. (**C**) *Tbx3* is expressed in the crest

*Figure 3 continued on next page*

*Figure 3 continued*

of the ventricular septum, which showed early activation (same heart as in *Figure 1H–I*). (D) *Tbx3* expressing myocardium of the dorsal atrioventricular canal was damaged when atrioventricular block was induced (dorsal cut n = 3 and ventral cut n = 3).

DOI: https://doi.org/10.7554/eLife.32120.007

The following figure supplements are available for figure 3:

**Figure supplement 1.** Expression of *Tbx3* and *Tbx5* were similar in embryos of American alligator and Cuvier's dwarf caiman, suggesting evolutionary conservation of expression between crocodilians.

DOI: https://doi.org/10.7554/eLife.32120.008

**Figure supplement 2.** Characterization of the ventricular septum in 1-year-old alligators.

DOI: https://doi.org/10.7554/eLife.32120.009

---

*Chiodi and Bortolami, 1967*). As mammals and birds evolved independently from ectothermic ancestors, the simplest yet perhaps unsatisfying conclusion is that their highly similar conduction systems evolved independently as well in relation to endothermy. However, crocodilians are ectotherms and have much lower heart rates than mammals and birds of comparable size and, unique among ectotherms, have a full ventricular septum, which in mammals and birds is the structural substrate of the atrioventricular bundle and branches. Our study shows the presence of a specialized atrioventricular conduction pathway within the broad myocardial connection between the atria and ventricles, its position corresponding to the mammalian and avian atrioventricular bundle. In contrast to the settings of mammals and birds, the crocodilian atrioventricular bundle is not fully insulated by connective tissues. Nevertheless, specific cuts through the *Tbx3*-positive bundle caused atrioventricular block, implicating the *Tbx3*-negative myocardial continuity between the atria and ventricles was not sufficient to maintain atrioventricular conduction. Thus, although not being fully insulated, the *Tbx3*-positive myocardial connection acts as an atrioventricular bundle, as it does in mammals and birds.

Century-old anatomical studies on alligators already described dorsal atrioventricular myocardium extending onto the ventricular septum (*Greil, 1903*) (*Supplementary file 1*) in a manner that much resembles the domain of *Tbx3*-expressing myocardium described here. Much later, *Christian and Grigg (1999)* used plunge electrodes to show in the freshwater crocodile that ventricular activation initiates in the top of the septum and propagates from there in a manner that we have also recorded in the American alligator (*Figure 1F*). The presence of a specialized atrioventricular conduction axis, the main finding of this manuscript, was not investigated in that study. Building on the previous findings, our results suggest that the evolution of the atrioventricular bundle predates that of endothermy and instead correlates with the formation of a full ventricular septum. It would be interesting to investigate whether other reptiles with partial ventricular septums, such as varanids, show signs of a *Tbx3*-positive atrioventricular bundle as well.

In mammals and birds, the atrioventricular bundle ramifies into bundle branches which express *Tbx3* and drape the left and right surface of the ventricular septum (*Hoogaars et al., 2004*). In the crocodilians, we never saw similar branches. In the freshwater crocodile, *Christian and Grigg (1999)* found a dorsal and a ventral 'rapid channel' within the ventricular septum, but the channels were not distinctive by histology. It is not clear whether these channels can be considered homologous of the bundle branches of mammals and birds. One anatomical study claimed the presence of bundle branches in crocodilians (*Mathur, 1971*), but the most prevalent view is that the crocodilian heart is without a histologically distinct atrioventricular bundle and bundle branches (*Davies et al., 1952*; *Christian and Grigg, 1999*; *Greil, 1903*; *Swett, 1923*).

The question remains to what extent the trabecular myocardium of the alligator can be equated to the mammalian and bird Purkinje network. In mammals, the embryonic trabecular myocardium gives rise to the Purkinje network that can be identified by the expression of key genetic markers *Cntn2*, *Nppa* and *Gja5* (Cx40) (*van Weerd and Christoffels, 2016*; *Miquerol et al., 2010*; *Pallante et al., 2010*). In the alligator, *Cntn2* is absent, *Nppa* is scattered rather than confined to the myocardium around the central lumen (*Jensen et al., 2017b*), and *Gja5* (Cx40) is expressed in the trabeculated and compact myocardium rather than confined to a miniscule subset of the ventricular myocytes. Therefore, the trabecular myocardium of the alligator is unlike the Purkinje system, but resembles that of other ectotherms. The activation of the ventricles of ectotherms relies on relatively slow conduction in trabecular myocardium instead of fast conduction through the atrioventricular

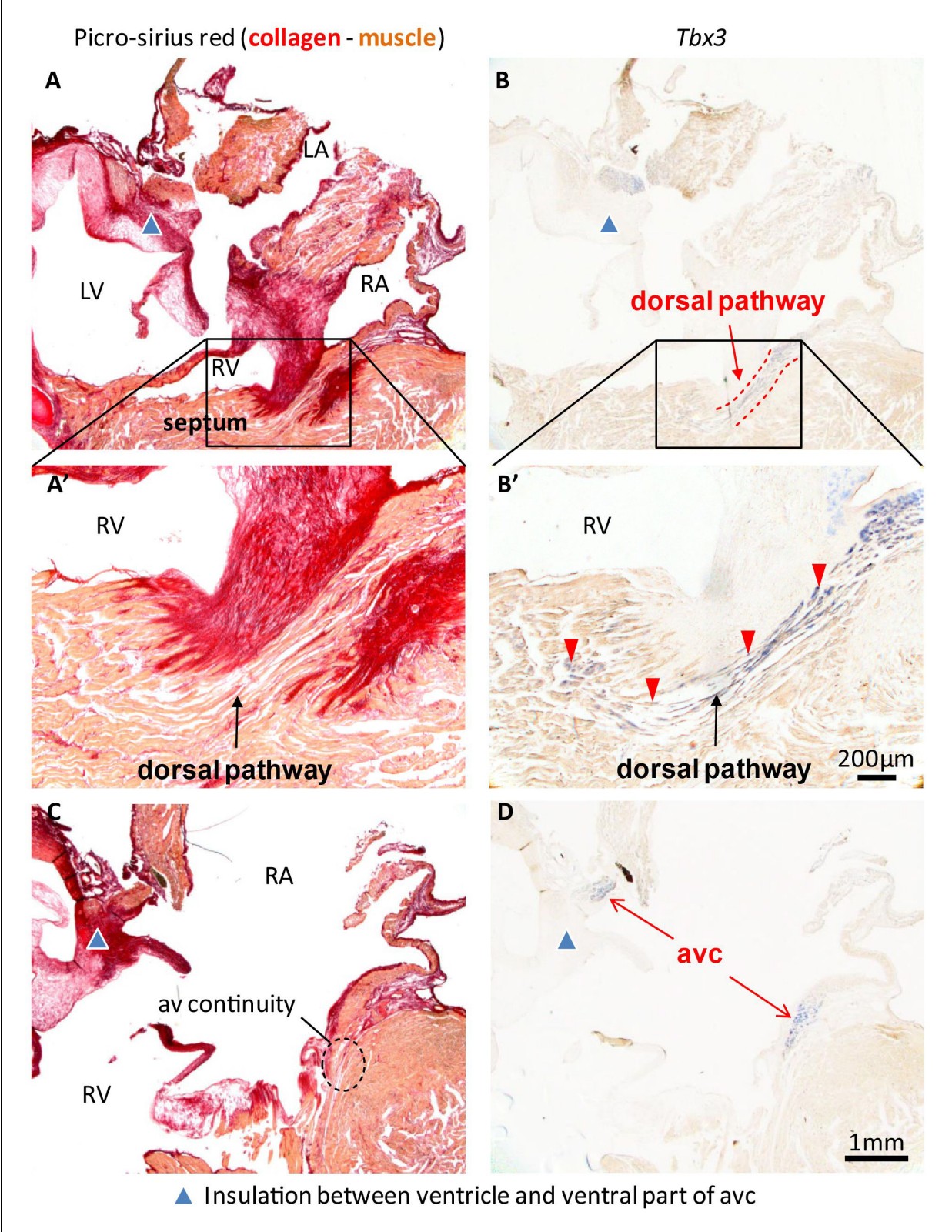

**Figure 4.** Insulation by collagen in the atrioventricular junction and near the atrioventricular bundle. (**A**) Sagittal section of the ventricular base of a 1-year alligator, showing complete insulation by connective tissue ventrally (blue arrowhead) and myocardial atrioventricular continuity dorsally (insert, enlarged in A'). (**B-B'**) The dorsal pathway expresses *Tbx3*. (**C**) Dorsally, there is broad atrioventricular continuity, this section is 800 μm to the right of
*Figure 4 continued on next page*

*Figure 4 continued*
the section of A and there is still atrioventricular continuity (dashed circle). (D) In this region of the heart, *Tbx3* is confined to the atrioventricular canal (avc) and is not expressed in the ventricle as in B.
DOI: https://doi.org/10.7554/eLife.32120.010

bundle branches and Purkinje system as seen in mammals and birds. Indeed, the QRS duration, a functional measure for Purkinje system, was substantially longer in alligators than in comparatively sized mammals, even when heated to mammalian body temperatures. In other crocodilians, the QRS duration is also approximately 100 ms suggesting that the speed of ventricular activation is similar across crocodilians (*Christian and Grigg, 1999*; *Davies et al., 1951*; *Heaton-Jones and King, 1994*; *Syme et al., 2002*; *Wang et al., 1991*). It further suggests that the specialized atrioventricular conduction pathway does not have an appreciable effect on ventricular activation time. Taken together, we conclude that the mammalian and avian ventricular Purkinje network evolved from the ancient trabeculated ventricle still present today in ectotherms, including crocodiles. This is in line with the unexpected finding that the cardiac natriuretic peptide locus, which specifically marks the trabecular layer, has remained unchanged during alligator evolution (*Figure 5*). Therefore, this locus has changed independently after the branching of birds within the archosaur group (to which also crocodilians belong), which lost *Nppa*, and of mammals, which lost *CNP-I* and other genes.

We interpret the biphasic upstroke in the alligator heart as resulting from early trabecular activation and late activation of the compact myocardium (*Figure 7A–B*, *Figure 7—figure supplement 1*). The reconstructed activation maps show that the recorded biphasic upstrokes are not the result of two subepicardial activation waves propagating next to each other (*Efimov et al., 1999*). Instead,

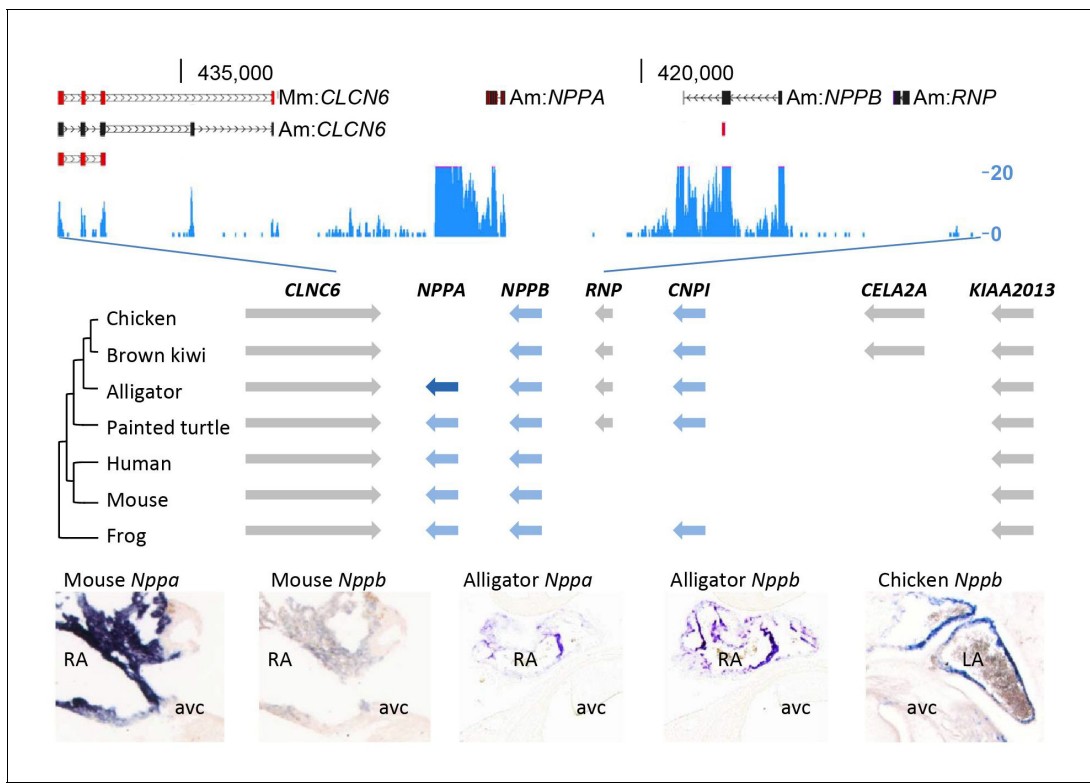

**Figure 5.** Evolution of the natriuretic peptide gene locus. RNA-sequencing tag counts aligned to BAC JH731559 revealed the unexpected presence of transcripts of orthologues of both *Nppa* and *Nppb*. The alligator cardiac natriuretic peptide gene locus is homologous to that of amphibians (frog) and reptiles (turtle). Birds have lost *Nppa,* and *CELA2A* and surrounding sequences have been inserted. Mammals have replaced the region in between *Nppb* and *KIAA2013*. The presence of *Nppa* and *Nppb* transcripts was validated by *in situ* hybridization using specific probes. Am, *Alligator mississippiensis*; avc, atrioventricular canal; Hs, *Homo sapiens*; LA, left atrium; Mm, *Mus musculus*; RA, right atrium.
DOI: https://doi.org/10.7554/eLife.32120.011

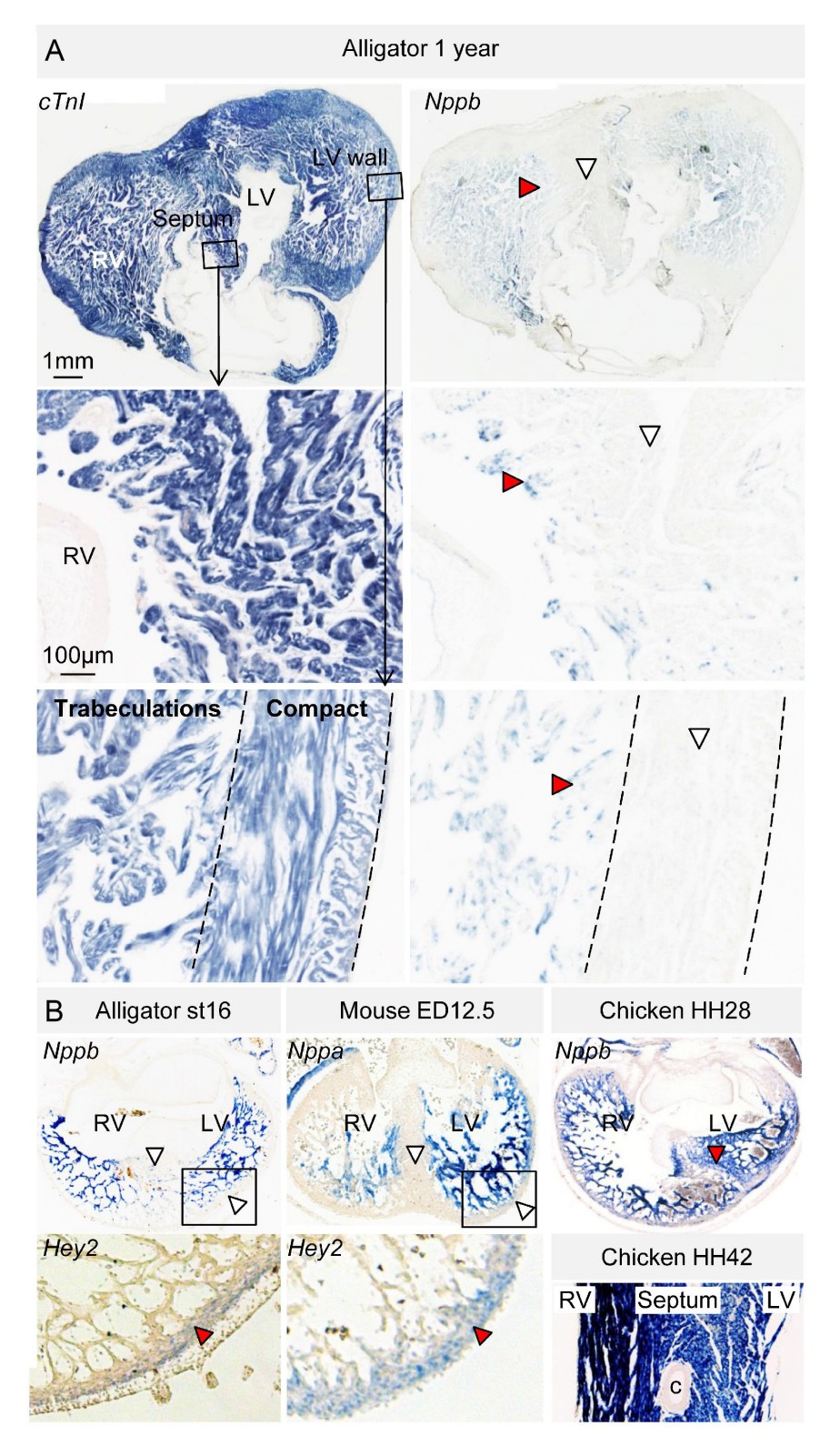

**Figure 6.** Evolutionary conservation of the trabeculated and compact ventricular wall. (**A**) The ventricular septum of the alligator is highly trabeculated (also *Figure 6—figure supplement 1*). Yet the alligator septum does not express the trabecular marker *Nppb* (n = 5). (**B**) The developing septum of alligators and mouse is empty for *Nppb*, in contrast to chicken (HH28 and HH42), whereas the developing septum of alligators and chicken is highly

*Figure 6 continued on next page*

*Figure 6 continued*

trabeculated, in contrast to the septum in mouse which is compact. The compact wall of both alligator (n = 1), caiman (n = 1), and mouse expresses *Hey2*. Arrowheads (white, no stain; red, stain); c, coronary artery; LV, left ventricle; RV, right ventricle.

DOI: https://doi.org/10.7554/eLife.32120.012

The following figure supplements are available for figure 6:

**Figure supplement 1.** Expression of *Nppa* and *Nppb* in the ventricular base of 1 year old alligators.

DOI: https://doi.org/10.7554/eLife.32120.013

**Figure supplement 2.** In the juvenile alligators, *Gja5* expression was found throughout the trabecular ventricle and in the substantial compact myocardium (more than 0.5 mm thick).

DOI: https://doi.org/10.7554/eLife.32120.014

each part of the upstroke belongs to a distinct activation wave from which separate activation maps can be reconstructed (*Figure 7—figure supplement 1*). Indeed, a large fraction of the excited and emitted light can traverse more than 1 mm of tissue (*Walton and Bernus, 2015*), which exceeds the thickness of the compact myocardium in the alligator hearts (290–860 µm). Of six hearts tested, only the two largest animals showed fractionated upstrokes, possibly because the smaller animals had a less thick compact wall, which did not provide sufficient emission light to overcome the noise. Thus, we conclude that in crocodiles first the atrioventricular bundle and trabecular myocardium activates, and subsequently the compacted myocardium, a sequence resembling that of mammalian and bird hearts where the atrioventricular bundle, its branches, and Purkinje network activate prior to the bulk of the compact wall. We speculate that the slowly conducting trabecular ventricle with a thin compact layer of fish and amphibians and the fast-conducting small Purkinje system with thick compact walls of mammals and birds represent two extreme situations of a spectrum. The alligator would be in-between those extremes. Investigation of the presence of bi-phasic upstrokes in other ectotherms with significant compact walls, such as some species of fish (*Icardo, 2017*; *Farrell and Smith, 2017*), may provide indications for such a spectrum. This would imply that precursor components of the Purkinje system were present before the evolution of endothermy and could stem from the earliest trabeculated ventricles as seen in jawless fish.

## Conclusion

We show the presence of a molecularly distinctive functional atrioventricular conduction system in the alligator heart. The presence of such a system in the crocodilian heart challenges our understanding of cardiac evolution. It suggests the development of the specialized atrioventricular conduction system predates the development of endothermy and correlates instead with the formation of a full ventricular septum. From this insight, one may be able to predict a relation between the degree of development of the ventricular septum and the presence of the atrioventricular bundle.

## Materials and methods

### Animals

The investigation conforms with the guide for the Care and Use of Laboratory Animals published by the US National Institutes of Health (NIH Publication No. 8523, revised 1996) and was approved by the Institutional Animal Studies Care and Use Committee of the University of North Texas (IACUC #1403–04). We used six female alligators (1.2 ± 0.4 kg, avg ±sd) to investigate the electrophysiology of the heart. The hearts of the six female alligators together with a series of embryonic alligators (n = 3, Ferguson stages 13, 16, 18), and embryonic Cuvier's dwarf caiman (*Paleosuchus palpebrosus*, n = 2, Ferguson stages 13 and 16) were used to molecularly characterize the heart. Ferguson (*Ferguson, 1985*) stages 13 and 18 correspond approximately to chicken Hamburger-Hamilton (*Hamburger and Hamilton, 1992*) stages 25 and 32, respectively, and mouse embryonic day 12 and 15.5, respectively. Additional four female alligators (2.7 ± 1.0 kg, avg ±sd) were used to study the in vivo effect of temperature on the QRS duration of the ECG.

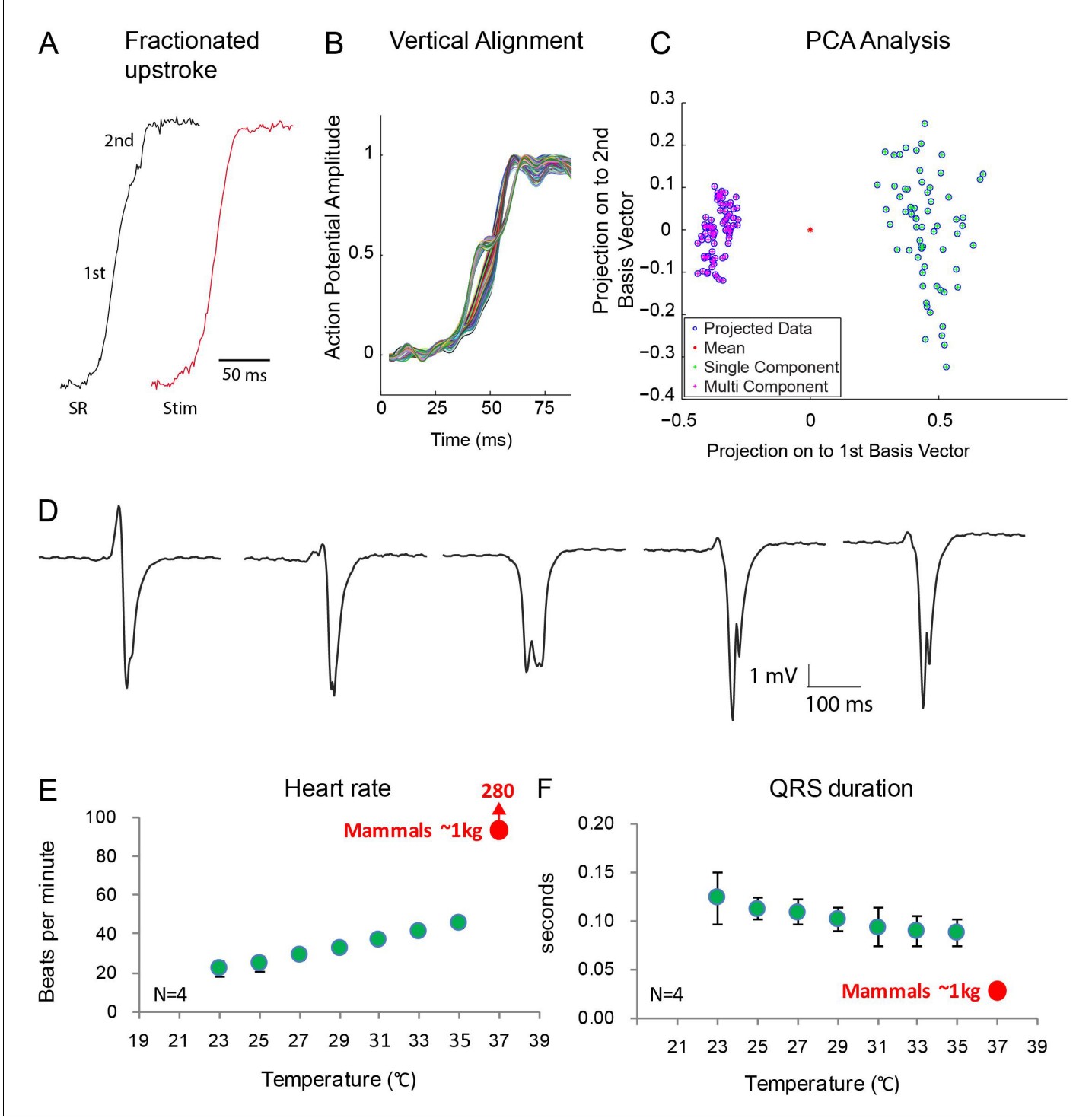

**Figure 7.** Fractionated upstroke of the optical action potentials indicated two distinct tissue layers. (**A**) Upstroke during sinus rhythm (SR) and induced by stimulation (Stim). (**B**) Vertical alignment of optical action potentials on the dorsal side of the alligator heart. (**C**) Principle component analysis (PCA) illustrating the different morphology of the upstroke with and without fractionation. (**D**) shows local electrograms recorded from the dorsal and ventral side of the heart shown in *Figure 7—figure supplement 1*. (**E-F**) *In vivo* heating of anaesthetized 2-year-old alligators leads to an increase in heart rate and a decrease in ventricular activation time as assessed by QRS duration (points are averages, error bars are standard deviation). The QRS duration of warm alligators never approach the much shorter QRS duration of eutherian mammals (*Detweiler, 2010*).

DOI: https://doi.org/10.7554/eLife.32120.015

The following figure supplement is available for figure 7:

*Figure 7 continued on next page*

*Figure 7 continued*

**Figure supplement 1.**
DOI: https://doi.org/10.7554/eLife.32120.016

## ECG recording *in vivo*

All four adult alligators were lightly anesthetized with Isoflurane. Electrodes were placed at the right (R) and left (L) side of the chest (ventral). ECG was recorded using PowerLab 26T (AD-Instruments, Colorado Springs, CO, sampling rate 2 kHz) and tracings were analyzed using Labchart Pro software. We calculated Lead I by subtracting L from R.

## *In vivo* effect of temperature on the QRS

Animals were fasted for 6 to 8 days prior to experimentation. On the day of study four alligators (mass = $2.7 \pm 1.0$ kg) were lightly anaesthetized by placing them in a sealed container with gauze soaked in isoflurane (Isoflo; Abbott laboratories, North Chicago, IL). The animal's trachea was then intubated and the lungs artificially ventilated using a ventilator (Model 552; Harvard Apparatus; Holliston, MA) downstream of a vaporizer (Ohmed Fluotec 4 Anesthetic Vaporizer; GE Healthcare; Buckinghamshire, UK) providing 2% isoflurane at 5–10 breaths $min^{-1}$. A modified air pump was used to continuously pull room air through the vaporizer. Once a surgical plane of anesthesia was achieved the animal was placed ventral side up on a heat pad (Homeothermic Blanket Control Unit, Harvard Apparatus; Holliston, MA). In addition, a lamp with a 40-watt bulb was placed above the animal at approximately the midpoint of the thoracic cavity. Three 12 mm surgical steel needle electrodes (29 gauge) were inserted subcutaneously in the animal. Two electrodes were place approximately 2 cm below the posterior end of the sternum one on left and right lateral side of the body wall. A third electrode was placed posteriorly at the intersection of the right rear limb and the body wall. Each electrode was connected to an amplifier (Animal Bio Amp, Adinstruments Colorado Springs, CO) connected to a PowerLab data recording system (4 ST Adinstruments Colorado Springs, CO, USA) connected to a Macintosh computer running Chart software (Chart 8 Adinstruments Colorado Springs, CO). Data was collected as a sampling rate of 1000 Hz with a low pass (1 Hz) and a high-pass (50 Hz) filter setting. Once electrodes were in place a thermocouple was advanced approximately 3 cm into the cloaca and connected to a temperature meter (BAT-12, Physitemp Instruments, Clifton, NJ). Output for the temperature meter was connected to the PowerLab data recording system. Once the instrumentation was completed the thermal blanket and lamp were turned on. Animal core body temperature was then increased gradually from ~21.5°C to 36.0°C. At the completion of the study animals were euthanized with an intravenous injection of pentobarbital (350 mg/kg).

## Optical and electrical recordings *ex vivo*

For the *ex vivo* experiments, we used seven juvenile alligators ($\pm1.3$ kg). First, we recorded and ECG from anesthetized animals (similar to described above) after which the animals were put in deep anesthesia by an injection of 50 mg $kg^{-1}$ body mass pentobarbital in the pre-cranial venous sinus. Subsequently the hearts were excised through a ventral incision. The excised hearts were placed in a bath of a custom-made Ringer solution at room temperature (22° Celcius) and perfused through PE90 catheters inserted in the ventral wall of the left and right atria (Ringer, in mmol/l: NaCl 120, Tris 5, $NaH_2PO_4$ 1, KCl 2.5, $MgSO_4$ 1, $CaCl_2$ 1.5, Glucose 5, pH adjusted to 7.5 with HCl). From the excised hearts, we recorded a pseudo-ECG (*ex vivo*) from electrodes placed in the tissue bath, 5 mm from the heart (Powerlab 26T; AD-Instruments, Colorado Springs, CO).

For recording of optical action potentials, we first loaded the hearts by superfusion with 10–20 μM di-4-ANEPPS (Molecular Probes, Eugene, OR) for 10 min which gave a preferentially loading the sub-epicardial compact wall of the ventricular myocardium. After loading, the excitation-contraction uncoupler blebbistatin (10–100 μM, Tocris Bioscience, Ellisville, MO) was added to the perfusate but failed to remove motion artifacts. Excitation light was delivered by a $520 \pm 5$ nm light emitting diode (Prizmatix, Southfield, MI) and emitted fluorescence was filtered >610 nm and recorded by a CMOS-sensor, (100 × 100 elements, 1 kHz, MICAM Ultima, SciMedia Ltd, Costa Mesa CA). One heart ceased spontaneous beating and was excluded from further experiments. After the first measurement, 5 out of the remaining six hearts were loaded for a second time by perfusion of 10–20 μM di-

4-ANEPPS to load the trabeculations of the ventricular lumen. Local electrograms were recorded from the epicardium of the ventral and dorsal side using a Franz electrode. In the bath, a ECG was recorded between two electrodes placed at the 2.5 cm from the left and right side of the heart. In three hearts, we stimulated on the apex (dorsal) at 0.5 Hz. Optical signals were analyzed using custom made Matlab-based software (*Laughner et al., 2012*) (http://efimovlab.org/content/rhythm). Principle component analysis was performed using Matlab2014. The optical mapping of the alligator was compared to optical mappings of *Xenopus* and *Anolis* that were re-analyzed from published findings (*Jensen et al., 2012*).

## Induced atrioventricular block by surgical incisions

In three hearts, we cut the atrial and ventricular tissue in the vicinity of the crux, that is, the intersection of the atrioventricular sulcus and the dorsal descending coronary artery, which reveals the approximate position of the atrioventricular node in the hearts of mammals (*Crick et al., 1998*). In the other three hearts, we cut the ventral and lateral parts of the atrioventricular junction. After the experiments, the hearts were fixed for 24 hr in 4% paraformaldehyde in phosphate buffered saline (adjusted to pH 7.5 with NaOH) and then kept in 70% ethanol.

## Sectioning, in situ hybridization, and histological stains

The ventricular base including atrial tissue was dissected free and embedded in paraffin and sectioned in 10 or 12 µm thick sections in either the frontal, transverse, or sagittal plane. We performed in situ hybridization as previously described (*Moorman et al., 2001*). Probes for American alligator mRNA transcripts were made based on the following coordinates using UCSC Genome Browser on American alligator Aug. 2012 (allMis0.2/allMis1) Assembly: *Cntn2* (JH737142:190,064–195,365), *cTnI* (JH732499:9,192–12,570), *Gja5* (JH733970:656,237–659,259), *Hey2* (JH734274:510,337–510,783), *Nppa* (JH731559:424,483–424,843), *Nppb* (JH731559:415,415–417,373), *Scn5a* (JH739807:162,160–168,550), *Tbx3* (JH733970:656,237–659,259), *Tbx5* (JH733970:893,979–894,506). Probes for *Anolis* mRNA transcripts were based on UCSC Genome Browser on Lizard May 2010 (Broad AnoCar2.0/anoCar2) Assembly; *Myh6* (chrUn_GL343680:156,017–156,398). To detect collagen on tissue sections, we incubated sections for more than one hour in picro-sirius red followed by 2 min differentiation in 0.01M HCl.

## RNA isolation and sequencing

Tissue samples were collected and homogenized with a homogenizer (Ultra-Turrax, IKA, DE) in lysis buffer RA1 of the NucleoSpin total RNA isolation kit (RNA II, Macherey-Nagel, DE). RNA isolation was performed following manufacturer's protocol. RNA purity, concentration and integrity were determined by respectively Nanodrop (ND-1000, Isogen Life Science, NL), Qubit RNA Broad-Range (2.0, Life Technologies), and Bioanalyzer RNA Nano (2100, Agilent Technologies). Amplified double stranded cDNA was generated using the Ovation RNA-Seq system (V2, Nugen) following manufacturer's protocol with 32 ng RNA input per sample. cDNA purity, concentration and size distribution were determined by Nanodrop, Qubit DNA HS and Bioanalyzer DNA 1000. Libraries were made using the 5500 SOLiD Fragment Library Core kit (Life technologies) and size distribution was determined by Bioanalyzer DNA 1000. RNA sequencing was performed on the SOLiD 5500 Wildfire (Life technologies). Sequence tags were mapped with Lifescope on the AllMis1 Aug. 2012 contig assembly of the American alligator genome.

## Statistics

All analyses were performed using IBM SPSS Statistics 24. Statistical significance between *in vivo* and *ex vivo* derived ECG parameters was determined using a paired sample t-test. Other group comparisons were performed using one-way ANOVA with Bonferroni test for post hoc matched pairs. Sample size (n) is given in each figure legend. Expression patterns in histological sections were established based on section from at least two different animals. Values are given as mean ±SEM. We considered a p-value lower than 0.05 as statistically significant.

## Acknowledgements

We thank Loren J Field and Antoon Moorman for helpful discussions, Vincent Wakker for generating RNA probes and John Eme for collection of alligator eggs.

## Additional information

### Funding

| Funder | Grant reference number | Author |
|---|---|---|
| Carlsbergfondet | | Bjarke Jensen |
| Fondation Leducq | | Vincent M Christoffels |
| CVON HUSTCARE | | Vincent M Christoffels |
| Hartstichting | COBRA3 | Vincent M Christoffels |
| Hartstichting | 2016T047 | Bastiaan J Boukens |
| Hartstichting | CONCOR-genes, CVON2014-2018 | Alex V Postma Vincent M Christoffels |
| Czech Science Foundation | 16-02972S | David Sedmera |

The funders had no role in study design, data collection and interpretation, or the decision to submit the work for publication.

### Author contributions

Bjarke Jensen, Conceptualization, Resources, Data curation, Formal analysis, Funding acquisition, Investigation, Visualization, Methodology, Writing—original draft, Project administration, Writing—review and editing; Bastiaan J Boukens, Conceptualization, Data curation, Formal analysis, Investigation, Visualization, Methodology, Writing—original draft, Project administration, Writing—review and editing; Dane A Crossley II, Resources, Data curation, Methodology, Project administration, Writing—review and editing; Justin Conner, Rajiv A Mohan, Data curation, Investigation, Writing—review and editing; Karel van Duijvenboden, Data curation, Formal analysis, Writing—review and editing; Alex V Postma, Formal analysis, Methodology, Writing—review and editing; Christopher R Gloschat, Formal analysis, Investigation, Writing—review and editing; Ruth M Elsey, David Sedmera, Resources, Writing—review and editing; Igor R Efimov, Resources, Funding acquisition, Writing—review and editing; Vincent M Christoffels, Conceptualization, Resources, Formal analysis, Investigation, Writing—original draft, Project administration, Writing—review and editing

### Author ORCIDs

Bjarke Jensen  https://orcid.org/0000-0002-7750-8035
Rajiv A Mohan  https://orcid.org/0000-0002-3622-1759
Igor R Efimov  http://orcid.org/0000-0002-1483-5039
Vincent M Christoffels  http://orcid.org/0000-0003-4131-2636

### Ethics

Animal experimentation: The investigation conforms with the guide for the Care and Use of Laboratory Animals published by the US National Institutes of Health (NIH Publication No. 8523, revised 1996) and was approved by the Institutional Animal Studies Care and Use Committee of the University of North Texas (IACUC #1403-04).

### Decision letter and Author response

Decision letter https://doi.org/10.7554/eLife.32120.024
Author response https://doi.org/10.7554/eLife.32120.025

## Additional files

### Supplementary files

• Source data 1. Contains three sheets. Sheet '*Figure 1E*' gives the parameters of the ECGs of the hearts used for optical mapping and surgical cuts. Sheet '*In situ* hybridization' provides an overview of all *in situ* hybridization and in which figures the data is shown. Sheet '*Figure 7E, F*' gives the parameters of the ECGs of the animals that were heated to mammalian body temperatures, summarized in *Figure 7E, F*.

DOI: https://doi.org/10.7554/eLife.32120.017

• Supplementary file 1. Supplementary Table 1: Overview of the literature. The presence of a specialized conduction system has been investigated in many reptile species and no consensus has emerged. The lack of consensus reflects the very heterogeneous quality of the previous studies and different definitions of 'specialization'. To claim specialization, anatomists placed much emphasis on how pale cells were, that is how Purkinje cell-like they were, whereas many electrophysiologists placed emphasis on function, for instance whether there was an atrioventricular delay under the influence of nervous activity. Currently, 'specialization' is much informed by molecular biological data in the setting of the mammal heart.

DOI: https://doi.org/10.7554/eLife.32120.018

• Transparent reporting form

DOI: https://doi.org/10.7554/eLife.32120.019

### Major datasets

The following dataset was generated:

| Author(s) | Year | Dataset title | Dataset URL | Database, license, and accessibility information |
|---|---|---|---|---|
| Jensen B | 2018 | Alligator mississippiensis Transcriptome or Gene expression | https://www.ncbi.nlm. nih.gov/bioproject/ PRJNA392860 | Publicly available at NCBI BioProject (Accession no. PRJNA392860) |

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
