## [Decision Letter]

Thank you for submitting your article "Specialized impulse conduction pathway in the alligator heart" for consideration by *eLife*. Your article has been reviewed by three peer reviewers, and the evaluation has been overseen by a Reviewing Editor and by Didier Stainier as the Senior Editor. The following individual involved in review of your submission has agreed to reveal his identity: Tobias Wang (Reviewer #2).

The reviewers have discussed the reviews with one another and the Reviewing Editor has drafted this decision to help you prepare a revised submission.

Summary:

In this manuscript, Jensen at el. examine atrioventricular conduction in the hearts of crocodiles, a rare example of an ectothermic vertebrate species that contains a complete ventricular septum. Using an appropriate (and impressive) array of experimental techniques, the authors demonstrate that the crocodilian heart has a specialized atrioventricular bundle that provides for electrical signals to be conducted from the atria to ventricle in a timely fashion, insuring the necessary delay required for blood to be transmitted between these chambers. Through histology and gene expression analysis, the authors establish that the AV bundle is found immediately above the ventricular septum, and its location is verified by ablation studies that show complete inhibition of electrical propagation upon mechanical lesion of the relevant area. Thus, the authors conclude that an AV conduction apparatus, similar to the His-bundle of mammals, exists within the crocodile heart. Since crocodiles display a slow heart rate, this argues that the atrioventricular bundle did not evolve in response to the high heart rates that attended the evolution of endothermy amongst the closely related birds (although this has been the dominant view for the past decade). This is an important conclusion with a significant impact on our view of the evolution of the cardiac conduction system in vertebrates. However, several points remain to be addressed in order to support the authors' conclusions more fully.

Essential revisions:

1) In mammals, the AV bundle is a specialized myocardial bridge that is electronically insulated and connects the AV node to the ventricular Purkinje system. The authors have concluded that the TBX3-positive myocardium running through the ventricular septum of the crocodile denotes a specialized atrioventricular conduction pathway. However, a similar TBX3-positive region is noted in the dwarf caiman which seems to lack a defined atrioventricular conduction pathway. The authors also note that no insulation is present around the crocodile AV-conducting pathway, which should allow for direct electrical communication with any adjacent myocardial population. This leads to two major concerns that the authors should address. First, could the authors more clearly articulate the criteria with which they would define a specialized AV bundle? Is it simply the myocardial continuity present between atria and ventricle, or does it need to have specialized conduction characteristics? Second, it is difficult to resolve whether the key evolutionary feature for AV conduction in the crocodile heart is that it has a ventricular septum which serves as the substrate for AV conduction, or that it contains components similar to an annulus fibrosus that can prevent broader AV conduction and restrict the region of AV myocardium capable of transmitting an impulse from atrium to ventricle. Could the authors clarify their thoughts on this issue? Also, could the authors provide data supporting their statement in the discussion that the crocodilian atrioventricular bundle is not insulated? Can the authors shed light on how the crocodilian heart avoids an inadvertent spread of the atrial depolarization outside the bundle?

2) In mammals, the atrioventricular bundle and Purkinje fiber network are connected by right and left bundle branches. It is unclear in this study whether these latter structures would have evolved with the atrioventricular bundle or with the Purkinje fiber network. This topic should be investigated and incorporated into the manuscript. Is there any histological evidence in favor of structures like bundle branches being present in the crocodile? How do the authors envisage that the activation wave is propagated to the ventral ventricular apex? How would the authors compare the scenario in the crocodile to the fast conducting network seen in the zebrafish ventricle, which seems to include base-to-apex propagation? What do the authors see as the functional benefit of the AV bundle in a heart with low frequencies of contraction?

3) On a somewhat related topic, could the authors include data from additional species to reinforce their broad evolutionary conclusions based on analysis of (albeit difficult to obtain) crocodile and caiman material? Discussion of data from the literature would be fine in this regard, if these direct comparisons can strengthen the evolutionary conclusion.

4) The authors use the biphasic nature of the electrogram upstroke as a demonstration of potential multilayer conduction. It is important to include the details of this analysis. What were the parameters used to separate the components of the upstroke? Can the derivative of these upstrokes be separated to demonstrate the actual temporal difference between inner and outer activation? How does this vary between where the surface myocardium overlays the septum vs where it overlays the trabeculae? Are similar biphasic electrograms present in other adult organisms lacking septum (fish, frog, lizards)? In addition, it would be helpful for the authors to indicate from which regions the provided upstroke recordings were taken.

5) The authors conclude that the epicardial breakthrough in the crocodile heart is towards the apex of the heart, which would be indicative of a conduction pathway through the ventricular septum/trabeculae. However, the data provided in Figure 7—figure supplement 1 seems to indicate epicardial breakthrough first occurs closer to the atria. Could the authors clarify this point?

[Editors' note: further revisions were requested prior to acceptance, as described below.]

Thank you for resubmitting your work entitled "Specialized impulse conduction pathway in the alligator heart" for further consideration at *eLife*. Your revised article has been favorably evaluated by Didier Stainier as the Senior Editor, Deborah Yelon as the Reviewing editor, and three reviewers.

All three reviewers were pleased with the revisions made to your manuscript, and two out of the three reviewers were completely satisfied with the revised version. One reviewer highlighted a few points in the text that you could clarify a bit further in order to make your interesting work as accessible as possible for the readers of *eLife*.

The comments of all three reviewers are included below. Could you please take a look at the comments from reviewer #3 (especially comments #1 and #2) to see if there are any textual adjustments that could help to avoid similar confusion from future readers of your work? Once you've provided us with any changes that you'd like to make in that regard, we will be happy to move forward with accepting your manuscript for publication.

*Reviewer #1:*

The authors have satisfactorily addressed my concerns. This study makes an interesting evolutionary point about the separate origin of distinct components of the ventricular conduction system relevant to its modular development in mammals.

*Reviewer #2:*

The authors have revised the manuscript in accordance with the comments/suggestions provided in the previous round of review. I have no additional comments or queries.

*Reviewer #3:*

The resubmitted manuscript by Jensen et al. revises a previous text in which the presence of a specialized atrioventricular conduction pathway is claimed to exist in the alligator heart. It was noted in the previous draft that the authors had generated an interesting panel of data, indicating the presence of conduction pathway along the crest of the ventricular septum, however, several concerns were raised. The authors have provided some additional information and modified the text in response to these concerns. In general, the newly incorporated data and text clarify some of issues raised by the reviewers, resulting in an improvement to the study. A few points still remain confusing.

1) Based on the initial manuscript, the impression given was that *Alligator mississippiensis* was selected for the study as "the only ectothermic vertebrates with a full ventricular septum." The response provided by the reviewers to point "1" seems to indicate that the dwarf caiman might also process a specialized AV pathway (in the absence of a septum). Is this a misreading of the response, or are the authors saying that a ventricular septum is in fact not required for and AV bundle? The authors have also now noted that the QRS for the American alligator is largely the same as QRS times recorded in ectotherms species that presumably lack a full septum (response to concern 3). This is again confusing; does this indicate that the AV bundle reported in this manuscript has little function consequence on ventricular conduction? If so, this should be emphasized in the text.

2) Regarding whether the myocardial bridge across the AV meets the criteria of a specialized conduction system: in fairness to previous work, the authors should state that they have used different criteria then historical work on this topic. The authors have defined "specialized" to mean the expression of TBX3 and the point of ventricular activation. As these are not the lineage criteria designated by earlier anatomists, this new definition should be clearly stated.

3) Given the lack of insulation, the lack of bundle branches, and how slow internal conduction in Figure 1I is, the mechanisms by which impulses would preferentially propagate down the septum and secondarily activate the compact layer remains difficult to understand.

4) Sample numbers and statistics related to physiological data remain a concern.

---

## [Author Response]

Essential revisions:1) In mammals, the AV bundle is a specialized myocardial bridge that is electronically insulated and connects the AV node to the ventricular Purkinje system. The authors have concluded that the TBX3-positive myocardium running through the ventricular septum of the crocodile denotes a specialized atrioventricular conduction pathway. However, a similar TBX3-positive region is noted in the dwarf caiman which seems to lack a defined atrioventricular conduction pathway.

We have now reconstructed the *Tbx3* positive domain of one more heart (in Figure 3—figure supplement 1), which allows the comparison of the American alligator and Cuvier’s dwarf caiman, and it shows that both species have a dorsal tongue of *Tbx3* that contacts the crest of the ventricular septum. We refer to Figure 3—figure supplement 1 in the subsection “Molecular characterization of the atrioventricular conduction pathway in alligator hearts”. For Cuvier’s dwarf caiman, we only have the expression pattern and cannot confirm by electrophysiology whether this *Tbx3* positive domain functions as a specialized pathway. Given the pronounced similarity between the American alligator and Cuvier’s dwarf caiman, we assume they have a similar specialized pathway.

The authors also note that no insulation is present around the crocodile AV-conducting pathway, which should allow for direct electrical communication with any adjacent myocardial population. This leads to two major concerns that the authors should address. First, could the authors more clearly articulate the criteria with which they would define a specialized AV bundle? Is it simply the myocardial continuity present between atria and ventricle, or does it need to have specialized conduction characteristics?

The presence of a specialized conduction system has been investigated in many reptile species and no consensus has emerged. The lack of consensus reflects the very heterogeneous quality of the previous studies and different definitions of “specialization” (see also de Haan, Circulation, 1961). To claim specialization, anatomists placed much emphasis on how pale (glycogen rich, fewer sarcomeres) cells are, that is how Purkinje cell-like they are, whether the tissue could be discriminated from the surrounding muscle mass by insulating connective tissue, etc., whereas many electrophysiologists placed emphasis on function, for instance whether there is an atrioventricular delay under the influence of nervous activity. Currently, “specialization” is much informed by molecular data. The criteria for specialization used in this study is twofold:1) Specific location of preferential conduction as indicated by incisions and 2) the presence of *Tbx3*, a highly conserved transcription factor that is the main marker of the developing and mature AV bundle in mammals and birds (Hoogaars, Cardiovasc Res. 2004; Bakker, Circulation Research 2008). To address this matter we have added the following to the Introduction section:

“[…]specialized atrioventricular conduction system tissues composed of distinctive cardiomyocytes (e.g. glycogen rich, less developed contractile apparatus, highly conductive) that can be morphologically distinguished from the working myocardium.”

Second, it is difficult to resolve whether the key evolutionary feature for AV conduction in the crocodile heart is that it has a ventricular septum which serves as the substrate for AV conduction, or that it contains components similar to an annulus fibrosus that can prevent broader AV conduction and restrict the region of AV myocardium capable of transmitting an impulse from atrium to ventricle. Could the authors clarify their thoughts on this issue? Also, could the authors provide data supporting their statement in the discussion that the crocodilian atrioventricular bundle is not insulated? Can the authors shed light on how the crocodilian heart avoids an inadvertent spread of the atrial depolarization outside the bundle?

We have performed additional histological analyses of insulating tissue (collagen by picro-sirius red) combined with detection of *Tbx3* on neighboring sections, and added a new figure (Figure 3). This shows connective tissue ventral and dorsal to the AV-conducting pathway identify by the expression of *Tbx3*. However, on the left and right of the dorsal *Tbx3* defined pathway, there is myocardial continuity between the atria and ventricles that does not express Tbx3. Therefore, the AV-conducting pathway is not fully insulated as it is in mammals and birds.

It is unclear whether inadvertent spread of the atrial depolarization outside the bundle occurs. However, specific cuts through the *Tbx3* positive bundle caused AV block, implicating that possible conduction through the *Tbx3* negative myocardial continuity was not sufficient to maintain AV conduction. The latter indicates that, although not being fully insulated, the *Tbx3* positive myocardial connection acts as an AV bundle. We have added the following text to the manuscript:

Results:

“Dorsally, the atrioventricular canal myocardium was nestled between collagen of the atrioventricular valves and the atrioventricular sulcus (Figure 3). The Tbx3 identified atrioventricular bundle extended from this sheet of atrioventricular canal myocardium (Figure 3). Laterally, the Tbx3 identified atrioventricular bundle was not insulated by connective tissue in contrast to the setting in mammals and birds.”

Discussion:

“Nevertheless, specific cuts through the Tbx3 positive bundle caused atrioventricular block, implicating the Tbx3-negative myocardial continuity between the atria and ventricles was not sufficient to maintain atrioventricular conduction. Thus, although not being fully insulated, the Tbx3 positive myocardial connection acts as an atrioventricular bundle, as it does in mammals and birds.”

2) In mammals, the atrioventricular bundle and Purkinje fiber network are connected by right and left bundle branches. It is unclear in this study whether these latter structures would have evolved with the atrioventricular bundle or with the Purkinje fiber network. This topic should be investigated and incorporated into the manuscript. Is there any histological evidence in favor of structures like bundle branches being present in the crocodile?

The proximal parts of the mammalian and bird bundle branches specifically express Tbx3 and other markers. We never found similar branches in the crocodilians, neither with histology nor *in situ* hybridization for *Tbx3, Tbx5, Gja5*, and *Scn5a*. We now make this point explicitly in Results and Discussion.

Results:

“In mammals and birds, the atrioventricular bundle and Purkinje network are connected by the Tbx3-expressing bundle branches on the left and right flanks of the ventricular septum. We never observed similar branches in the crocodilians, neither with histology nor in situ hybridization for Tbx3, Tbx5, Gja5, and Scn5a.”

Discussion:

“In mammals and birds, the atrioventricular bundle ramifies into bundle branches which express Tbx3 and drape the left and right surface of the ventricular septum (Hoogaars et al., 2004). […] One anatomical study claimed the presence of bundle branches in crocodilians (40), but the most prevalent view is that the crocodilian heart is without a histologically distinct atrioventricular bundle and bundle branches (Davies, Francis and King, 1952; Christian and Grigg, 1999; Greil, 1903; Swett, 1923).”

At least on this point there appears to be agreement in literature. (Concerning histological evidence, as reported in literature, Christian and Grigg (1999) identified two “channels” of preferential conduction by plunge electrodes within the crocodilian ventricular septum, but could not identify these with histology. Greil (1903) and Swett (1923) observed that atrioventricular canal-like myocardium projected onto the ventricular septum in the American alligator. However, they did not find bundle branches. Davies et al. (1952) did not find any evidence for specialized tissues in the American alligator. Importantly, Davies et al. are the same investigators who provided major contributions to the description of the specialized conduction system of mammals and birds. Only, Mathur (1971), who studied two species of crocodile, claimed to have found an atrioventricular node and atrioventricular bundle – and further claimed these to be homologous to those of mammals and birds – but he did not claim to have found any bundle branches).

How do the authors envisage that the activation wave is propagated to the ventral ventricular apex?

We envision early propagation predominantly, if not exclusively, within the septum and the trabeculated myocardium. The earliest activation of the compact myocardium is dorsal and where the septum fuses with the compact wall. On the ventral side, the compact myocardium is activated later, via the apex.

How would the authors compare the scenario in the crocodile to the fast conducting network seen in the zebrafish ventricle, which seems to include base-to-apex propagation?

The ventricular muscle of all vertebrates propagates the electrical impulse much faster than the atrioventricular junctional muscle, but our heating experiment, shown in Figure 6, shows that the alligator ventricle has a larger activation time (QRS duration) than the ventricles of mammals of comparable size. This strongly suggests that impulse propagation in the alligator is slower than in mammals. We do not know of similar heating experiments in zebrafish, but we would expect them to show a longer QRS duration than would be expected for a similarly sized mammal (obviously, adult zebrafish are much smaller than the smallest mammals so this is theoretical only). In fish of approximately 1kg, the QRS interval is as long as in alligators (trout, Comp Biochem Physiol A 145 (2006): 158–165; pike, Comp Biochem Physiol A 159 (2011) 39–45) or longer (African lungfish, Am J Physiol-Heart Circ Physiol 232 (1977): H24-H34). Alligators nonetheless show elements of the apex-to-base activation that characterizes mammals and birds. This unusual pattern of activation, for an ectotherm, is presumably explained to a large extent by the presence of the specialized Tbx3-positive pathway. Zebrafish appear without a comparable manner of activation, although the exact manner of activation of the zebrafish ventricle is a point of contention; please see the discussion in Jensen et al. 2012.

What do the authors see as the functional benefit of the AV bundle in a heart with low frequencies of contraction?

A particular structure does not necessarily have to have a functional benefit in itself, if its presence is related to a trait that has been selected for (Gould and Lewontin, 1979, Proc Roy Soc London B). We currently do not see an advantage to the AV bundle in the crocodilians, nor do we rule out an advantage, but for the time being, we are content with the notion that the AV bundle may simply relate to the presence of a full ventricular septum.

3) On a somewhat related topic, could the authors include data from additional species to reinforce their broad evolutionary conclusions based on analysis of (albeit difficult to obtain) crocodile and caiman material? Discussion of data from the literature would be fine in this regard, if these direct comparisons can strengthen the evolutionary conclusion.

Electrocardiograms have been recorded in a number of crocodilian species and the QRS duration of the electrocardiogram is indicative of whether there is a mammal-like specialized ventricular conduction system (this manuscript, Figure 6). The QRS durations we recorded from American alligators, approximately 100 ms, were of similar magnitude to reported previously for the American alligator (Heaton-Jones and King, 1994; Syme et al. 2002), the Chinese alligator (Wang et al., 1991), the saltwater crocodile (Christian & Grigg, 1999), and the Nile crocodile (Davies et al., 1951). As requested by the reviewer, we have discussed data from literature to strengthen the evolutionary conclusion. We comment on this in the Discussion section:

“In other crocodilians, the QRS duration is also approximately 100 ms suggesting that the speed of ventricular activation is similar across crocodilians (Christian and Grigg, 1952; Davies, Francis and King, 1951; Heaton-Jones and King, 1994; Syme, Gamperl and Jones, 2002; Wang et al., 1991).”

4) The authors use the biphasic nature of the electrogram upstroke as a demonstration of potential multilayer conduction. It is important to include the details of this analysis. What were the parameters used to separate the components of the upstroke? Can the derivative of these upstrokes be separated to demonstrate the actual temporal difference between inner and outer activation? How does this vary between where the surface myocardium overlays the septum vs where it overlays the trabeculae? Are similar biphasic electrograms present in other adult organisms lacking septum (fish, frog, lizards)? In addition, it would be helpful for the authors to indicate from which regions the provided upstroke recordings were taken.

To test whether biphasic upstrokes could be separated from upstrokes with one phase we first performed a principle component analysis. This is shown in the newly added Figure 7 of the revised manuscript. After that we separated the upstrokes with one phase from those with two phases based on the dV/dt. We have generated a figure (Author response image 1) showing multiple dV/dt max calculated from the signals with a biphasic upstroke. Distinct activation maps, based on the maximum positive derivative (Figure 7—figure supplement 1), could be generated from both the first and the second part of the fractionated upstrokes. The activation maps on the left of Figure 7—figure supplement 1 show the reconstructed activation patterns from the first part of the upstroke and the activation maps on the right of Figure 7—figure supplement 1 shows the reconstructed activation patterns from the second part of the upstroke. To explain this topic better we have added the following to the revised manuscript.

Results:

“Optical mapping analysis indicated the presence of two functional layers. In 2 out of 6 animals, we observed action potentials with fractionated (biphasic) upstrokes, in which an initial steep deflection was followed by a lessening of the inclination only to be followed by a second steep deflection (Figure 7A-C, Figure 7—figure supplement 1). […] Together with the gene expression data, these data suggest the presence of two layers, a trabecular myocardial layer and a compact layer, which are activated in subsequent order.”

**Author response image 1. respfig1:** Fractionated upstrokes measured from the epicardium. The traces in A and B shows the optically recorded action potential (upper) and its the derivative (lower). Panel C and D show magnifications of the red box from panel A and B respectively.

5) The authors conclude that the epicardial breakthrough in the crocodile heart is towards the apex of the heart, which would be indicative of a conduction pathway through the ventricular septum/trabeculae. However, the data provided in Figure 7—figure supplement 1 seems to indicate epicardial breakthrough first occurs closer to the atria. Could the authors clarify this point?

The activation map shown in Figure 7—figure supplement 1 shows at the left the pattern based on the first part of the upstroke and at the right the pattern based on the second part of the upstroke. The earliest breakthrough is based on the first part of the upstroke and observed in the mid of the ventricle (Figure 7—figure supplement 1 right top). The first activation based on the second part of the upstroke is, as noted by the reviewer, at the base of the heart close to the atria (Figure 7—figure supplement 1D left top). This breakthrough, however, occurred later than at the mid of the ventricle. We have addressed this issue in the revised manuscript:

“Distinct activation maps could be generated from both the first and the second part of the biphasic upstrokes showing first activation in the mid of the ventricle at the dorsal side.”

[Editors' note: further revisions were requested prior to acceptance, as described below.]

Reviewer #3:[…] 1) Based on the initial manuscript, the impression given was that Alligator mississippiensis was selected for the study as "the only ectothermic vertebrates with a full ventricular septum." The response provided by the reviewers to point "1" seems to indicate that the dwarf caiman might also process a specialized AV pathway (in the absence of a septum). Is this a misreading of the response, or are the authors saying that a ventricular septum is in fact not required for and AV bundle?

We have not made it sufficiently clear that the full septum is found in all crocodilians, which comprise 23 species of alligators, caimans (technically also alligators), crocodiles, and gharials. In the Introduction, we now write “Among ectotherms, only crocodilians (alligators, crocodiles, and gharials) have a full ventricular septum…”.

The authors have also now noted that the QRS for the American alligator is largely the same as QRS times recorded in ectotherms species that presumably lack a full septum (response to concern 3). This is again confusing; does this indicate that the AV bundle reported in this manuscript has little function consequence on ventricular conduction? If so, this should be emphasized in the text.

The QRS duration is mainly defined by the conduction system components downstream of the AV bundle (in mammals, the Purkinje fibre network) in combination with the conduction velocity in the working myocardium. We have now added the following sentence to the sixth paragraph of the Discussion, where the QRS duration is discussed: “It further suggests that the specialized atrioventricular conduction pathway does not have an appreciable effect on ventricular activation time.”

2) Regarding whether the myocardial bridge across the AV meets the criteria of a specialized conduction system: in fairness to previous work, the authors should state that they have used different criteria then historical work on this topic. The authors have defined "specialized" to mean the expression of TBX3 and the point of ventricular activation. As these are not the lineage criteria designated by earlier anatomists, this new definition should be clearly stated.

We have now added the following sentence to the last paragraph of the Introduction: “Because previous anatomical investigations led to contradictory interpretations (Supplementary file 1), we use complementary functional and molecular criteria to define “specialization.”

3) Given the lack of insulation, the lack of bundle branches, and how slow internal conduction in Figure 1I is, the mechanisms by which impulses would preferentially propagate down the septum and secondarily activate the compact layer remains difficult to understand.

Insulation is not necessarily an effect of fibro-fatty tissues alone, but can also be brought about by so-called source-sink mismatches and poor coupling between cells (Kléber, A.G. and Rudy, Y., 2004. Physiological reviews, 84(2), pp.431-488.). The propagation of the impulse from the specialized atrioventricular pathway to the ventricular surface is the least investigated part of our study. It remains difficult for us to assess the relative contribution of insulating tissues, source-sink mismatches, and cellular coupling, and we are therefore reluctant to speculate on this matter.

4) Sample numbers and statistics related to physiological data remain a concern.

The scale of our efforts reflects that American alligators are not readily available for experimentation. Nonetheless, the pattern of activation we report is fully consistent with the previous study of Christian and Grigg (1999) and the QRS durations we measured are similar to those reported previously (5 references cited in Discussion), strongly suggesting that our physiological data is sound.